# Striatal-Inoculation of α-Synuclein Preformed Fibrils Aggravated the Phenotypes of REM Sleep without Atonia in A53T BAC-SNCA Transgenic Mice

**DOI:** 10.3390/ijms232113390

**Published:** 2022-11-02

**Authors:** Shinya Okuda, Takeo Nakayama, Norihito Uemura, Rie Hikawa, Masashi Ikuno, Hodaka Yamakado, Haruhisa Inoue, Naoko Tachibana, Yu Hayashi, Ryosuke Takahashi, Naohiro Egawa

**Affiliations:** 1Department of Neurology, Kyoto University Graduate School of Medicine, Kyoto 606-8507, Japan; 2iPSC-Based Drug Discovery and Development Team, RIKEN BioResource Research Center, Kyoto 619-0237, Japan; 3Center for iPS Cell Research and Application (CiRA), Kyoto University, Kyoto 606-8507, Japan; 4Department of Neurology, Center for Sleep-Related Disorders, Kansai Electric Power Hospital, Osaka 553-0003, Japan; 5Division of Sleep Medicine, Kansai Electric Power Medical Research Institute, Osaka 553-0003, Japan; 6Department of Human Health Sciences, Kyoto University Graduate School of Medicine, Kyoto 606-8507, Japan

**Keywords:** Parkinson’s disease, α-synuclein, RBD, REM sleep, REM sleep without atonia

## Abstract

Accumulation of α-synuclein (α-syn) is the pathological hallmark of α-synucleinopathy. Rapid eye movement (REM) sleep behavior disorder (RBD) is a pivotal manifestation of α-synucleinopathy including Parkinson’s disease (PD). RBD is clinically confirmed by REM sleep without atonia (RWA) in polysomnography. To accurately characterize RWA preceding RBD and their underlying α-syn pathology, we inoculated α-syn preformed fibrils (PFFs) into the striatum of A53T human α-syn BAC transgenic (A53T BAC-SNCA Tg) mice which exhibit RBD-like phenotypes with RWA. RWA phenotypes were aggravated by PFFs-inoculation in A53T BAC-SNCA Tg mice at 1 month after inoculation, in which prominent α-syn pathology in the pedunculopontine nucleus (PPN) was observed. The intensity of RWA phenotype could be dependent on the severity of the underlying α-syn pathology.

## 1. Introduction

Synucleinopathy represented by Parkinson’s disease (PD) is an entity of neurodegenerative diseases, of which the defining pathological feature is the presence of Lewy bodies in the brain. Lewy bodies are predominantly composed of alpha synuclein (α-syn) aggregation. The hypothesis posed by Braak et al. that α-syn pathology can propagate is based on the distribution of α-syn pathology in the postmortem human brain [1], implying a close relationship between the presence of α-syn pathology in the neurons and the clinical manifestations related to the affected neural circuits.

From the aspect of neurophysiology, REM sleep behavior disorder (RBD), which is confirmed by the presence of REM sleep without atonia (RWA) in polysomnography [2], has been recognized as the highest diagnostic strength in the prodromal state of PD [3]. RWA could be an earlier abnormal finding than clinical RBD, as a harbinger of PD and the severity of RWA, could increase with disease progression [4,5,6], suggesting that RWA could be a crucial clinical biomarker in PD patients. The causal relationship between RWA in RBD and the exogenous load of α-syn aggregation is unclear.

Previous research has shown that A53T mutated SNCA bacterial artificial chromosome transgenic (A53T SNCA BAC-Tg) mice exhibited hyposmia and the RBD-like phenotype at five months of age [7,8]. Inoculation of preformed fibrils of α-syn (PFFs) into the striatum in A53T SNCA BAC-Tg mice increased α-syn pathology in the substantial nigra compacta (SNc) and induced minor motor dysfunctions with dopaminergic neuron loss [9], recapitulating motor phenotypes as PD model animals.

As a part of the Special Issue “Molecular Research on Neurodegenerative Diseases 3.0”, we here report the sleep phenotypes of non-motor manifestations in striatal α-syn PFFs-inoculated A53T SNCA BAC-Tg mice.

## 2. Results

We inoculated mouse α-syn PFFs into the left striatum of A53T SNCA BAC-Tg mice. We used wild-type (WT) mice as the genetic control (i.e., A53T SNCA non-transgenic) and inoculated phosphate-buffered saline (PBS) as the control of exogenous inoculation. We first investigated the sleep architectures of four groups of mice (i.e., PBS-inoculated WT mouse (WT:PBS); PFFs-inoculated WT mice (WT:PFFs); PBS-inoculated A53T transgenic mice (Tg:PBS); PFFs-inoculated A53T transgenic mice (Tg:PFFs)). We found that the ratio of REM sleep stage was significantly increased in Tg:PFFs mice compared to WT:PBS mice at 1 and 2 mpi (Figure 1A,B). This was not significant at 4 mpi (Appendix A).

To examine RWA in each mice group, we then measured the electromyography (EMG) activity of neck muscle during REM sleep by calculating integral EMG activity during REM sleep. Quantified neck EMG activity during REM sleep was increased in Tg:PFFs mice at 1 mpi compared with WT:PBS control mice (Figure 2). Tg:PFFs mice exhibited several clusters of twitches of their body during REM sleep under video-monitoring (Appendix A). This increased EMG activity during REM sleep was relatively high in Tg:PFFs mice compared to WT:PBS at 2 mpi and 4 mpi, but the differences were not significant (Figure 2, Appendix A).

To identify the underlying α-syn pathology induced the phenotypes of RWA, we immunostained the whole brain of Tg:PFFs by phosphorylated α-syn antibody and focused on the five major regions which exhibited remarkable α-syn aggregates: prefrontal cortex (PFC), anterior cingulate cortex (ACC), dentate gyrus (DG), hippocampus (HIP), and pedunculopontine nucleus (PPN) (Figure 3A). We found dynamic pathological change of α-syn aggregates in PPN, which was the most prominent at 1 mpi (Figure 3B). Only Tg:PFFs showed this remarkable pathology of α-syn aggregates in PPN among four groups of mice (Figure 3C). P-Syn aggregates were colocalized in ChAT-positive cells, suggesting the dominant presence of α-syn PFFs aggregates in cholinergic neurons in PPN (Figure 3D).

## 3. Discussion

Isolated RWA precedes RBD and worsens with PD progression, which imply the underlying progressive process of α-syn pathology in PD [10]. In this study, we have first demonstrated that the intrastriatal inoculation of α-syn PFFs could aggravate the RWA phenotype in the genetic RBD model mice.

Changes in striatal cholinergic interneurons activity have been observed early in genetic PD rodent models [11]. Intrastriatal inoculation of α-syn PFFs propagates among major central nervous system (CNS) pathways [12]. Our previous study has demonstrated that intrastriatal-inoculated PFFs spread to various regions of the brain such as the ACC, PFC, DG, SN, HIP, and PPN in A53T SNCA BAC-Tg mice, which manifested motor dysfunction over 4 mpi with the loss of dopaminergic neurons immunostained by anti-tyrosine hydroxylase (TH) antibody in the SN [9]. In this study, we found that the PFFs:Tg mice showed the phenotypes of altered sleep architecture of REM sleep and RWA phenotypes, suggesting dysfunction of the regulatory neural circuit related to REM sleep. A recent study has demonstrated that inoculation of α-syn PFFs into the sublaterodorsal (SLD) tegmental nucleus induced selective apoptosis of SLD neurons, leading to stationary RWA phenotypes during 3 mpi [13]. We could not observe α-syn aggregates in the SLD in our striatal α-syn PFFs-inoculated mice. Instead, we observed prominent α-syn aggregate pathology in the PPN which reduced over 4 mpi. A-syn aggregates were observed in the cholinergic neurons in the PPN, which could be susceptible to α-syn PFFs propagation [14]. Thus, we speculated that there could be a positive correlation between the intensity of α-syn aggregate pathology in PPN and the RWA phenotypes in striatal α-syn PFFs-inoculated mice.

There are several limitations in our study. First, we could not intervene the neural circuitry by modifying the function of the neuron to examine the causal effect between α-syn PFFs aggregated in the specific region in the brain and sleep condition. Transient firing of the cholinergic neuron in PPN has been reported to modulate both REM sleep and NREM sleep in mice [15,16]. Intrastriatal α-syn PFFs could perturb the firing of the projected neuron and the striatal synaptic plasticity [17], suggesting that α-syn PFFs could trigger transient dysfunction of PPN cholinergic neurons. A future approach to modulate PPN cholinergic neurons using optogenetic or designed receptors such as designer receptors exclusively activated by designer drugs (DREADD) could have the potential to revert the RBD-like phenotype [18]. Second, we have not examined the focal field potentials of PPN. Recording focal field potential using the deep brain electrode might elucidate the more precise causal relationship between α-syn aggregate pathologies and RWA phenotypes in striatal α-syn PFFs inoculation model mice. Third, we could not find significant differences in the sleep phenotypes between Tg:PBS and Tg:PFFs. Our study showed the significant differences in RWA phenotypes between Tg:PFFs and WT:PBS at 4 months old, which is earlier than the findings of a previous study (i.e., at 11 months old between Tg and WT mice) [7], suggesting that the exogenous load of a-syn PFFs could aggravate the RWA phenotypes. Fourth, we only used male mice for sleep analysis in the present study since RBD is generally more common in males than in females. The previous study analyzing WT mice inoculated with α-syn PFFs into the CNS reported that male PFFs-injected mice developed more severe α-syn pathology than female ones [19], while there was no significant difference in REM sleep amounts between the sexes [20]. Future research should be done on both sexes to understand the disease in future studies. Further study to focus on the regulatory system of the neural circuit related to REM sleep may lead to novel disease-modifying therapy against α-synucleinopathy.

## 4. Materials and Methods

### 4.1. Animals

All mice used in this study were handled in accordance with the national guidelines. All procedures performed in this study were approved by the Institutional Animal Care and Use Committee, Institute of Laboratory Animals Graduate School of Medicine, Kyoto University (22501). A53T BAC-SNCA transgenic mice were generated as previously described [7].

### 4.2. Preparation of Preformed Fibrils (PFFs) of α-Synuclein

PFFs were generated as previously reported [7,21,22,23], with modifications. *Escherichia coli* BL21 (DE3) cells (BioDynamics Laboratory, Tokyo, Japan) were transformed with pRK172, encoding the mouse α-syn cDNA. The expression of α-syn was induced by 0.1 mM isopropyl β-D-1-thiogalactopyranoside (IPTG) for 4 h. The cells were pelleted by centrifugation at 4000× *g* at 4 °C for 5 min and lysed with several freeze–thaw cycles and subsequent sonications. The lysate was boiled for 5 min and centrifugated at 20,400× *g* at 4 °C for 15 min. The supernatant was filtered by ion exchange using Q sepharose fast flow (GE Healthcare, Little Chalfont, UK). Purification of α-syn was performed by dialysis using the buffer (150 mM KCl, 50 mM Tris-HCl, pH 7.5) and by ultracentrifugation at 186,000× *g* at 4 °C for 20 min. The concentration of the protein was measured using a BCA Protein Assay kit (Thermo Fisher, Waltham, MA, USA). Purified α-syn was diluted in buffer containing 0.1% (*w*/*v*) NaN_3_ to 7 mg/mL and incubated at 37 °C in a shaking incubator (SI-300C; AS ONE, Osaka, Japan) at 1000 rpm for 10 days. PFFs were pelleted by ultracentrifugation at 186,000× *g* at 20 °C for 20 min and stocked at −80 °C before use.

### 4.3. Surgery for the Inoculation of PFFs and Implantation of EEG/EMG Electronodes

Male SNCA A53T BAC-Tg mice (*n* = 8) and their wild-type (WT) littermates (*n* = 8) between 2 and 3 months of age underwent two surgeries to inoculate with 5 μg of α-syn PFFs or sterile phosphate-buffered saline (PBS) and subcutaneously implant a head mount with electroencephalography (EEG) and electromyography (EMG) electronodes (Biotex, Kyoto, Japan) at 2 weeks after inoculation. Mice were anesthetized with isoflurane and placed in a stereotaxic frame to fix the body. The head skin was incised with a sterile scalpel. A single needle insertion (co-ordinate s: +0.2 mm relative to Bregma, +2.0 mm from midline) was performed for the inoculation, targeting the dorsal left striatum (+2.6 mm beneath the dura). Two leads for measuring EEGs were implanted epidurally. A handheld drill (0.5 mm micro drill bit) was used to make two burr holes in the skull: the first at +1.0 mm anterior and 1.5 mm lateral to Bregma, the second at +1.0 mm anterior and 1.5 mm lateral to Lambda. The burr holes were each fitted with a dental screw and an exposed portion of the two EEG leads were fixed around the dental screws. The head mount and the leads were fixed on the scalp with dental acrylic. Two EMG leads were placed in direct contact with the cervical trapezius muscle, 2 mm apart along the same bundle of the muscle fibers. The animals were individually housed and allowed 7 days to recover from two surgeries.

### 4.4. EEG/EMG Data Acquisition

All recordings took place in a designated area with minimal noise or other background disturbance. The animals were individually housed in a soundproof cabinet equipped with fans that allowed for proper ventilation and temperature (25 °C) and humidity (55%) and light timers to maintain a strict 12:12 h light/dark cycle. The animals were allowed to move freely with cables connected to the implemented electrodes to output the EEG/EMG signal in plastic cage. The data were collected at a sampling rate of 250 Hz and the signals were amplified through an amplifier and digitized with an analog-to-digital converter (BAS-8103P; Biotex) and an appropriate software program (Sleepsign; Kissei Comtec, Tokyo, Japan). Animals were allowed at least 4 days to adapt to the recording conditions prior to each session.

### 4.5. Analysis of EEG and EMG

One 24-h period (12-h light/dark cycle; day 5 out of 7 total days recorded) was automatically scored using Sleepsign Recorder^®^ software (Kissei Comtec) into 10-s epochs of wake, non-REM sleep or REM sleep. We recorded EEG, EMG and actigraph detected by infrared light sensors to determine the sleep stage of four groups of mice at 4–7 weeks post inoculation (1-month post inoculation, 1 mpi), 8–11 weeks (2-month post inoculation, 2 mpi) and over 20 weeks (4-month post inoculation, 4 mpi). During the EEG/EMG recording, the behaviors of the animals were monitored by cameras to observe their motor activity during sleep. Wakefulness as the ‘WAKE’ stage were characterized by actigraph counts over one or with high amplitude of EMG power. Non-REM sleep was characterized by a relative increase in integrated FFT delta power in EEG amplitude. REM sleep was characterized by a relative increase in FFT theta rate (over 70%) with a relative low amplitude of EMG power. Lastly, the EEG/EMG signal quality obtained with the system with video-monitoring was checked by a visual inspection to confidently classify vigilance states with minimal artifacts. The percentage of integrated EMG power relative to the average of integrated EMG during REM sleep was calculated to determine tonic activity of the muscle during REM sleep.

### 4.6. Immunohistochemistry

Immunohistochemistry was performed on 8 μm thick serial sections. Primary antibodies were used to detect phosphorylated a-Syn (ab51253, Abcam, Cambridge, UK, 1:5000) and ChAT (AB144p, Sigma-Aldrich, St. Louis, Mo, 1:100). Each section was stained using 3′-diaminobenzidine (DAB; Vector Laboratories, Burlingame, CA, USA). Fluorescent secondary antibodies were used to examine immunoreactivity in double-labeled neurons (Alexa-fluor 488 or 594, Invitrogen, Carlsbad, CA, USA). Images were captured on a DP71 digital camera connected to a BX51 microscope (Olympus, Tokyo, Japan) or BZ-X710 (Keyence, Osaka, Japan). Mapping of pSyn pathology was performed by tracing all visible immune-reactive inclusions at 20× magnification from sections at multiple rostrocaudal levels corresponding to approximately 0.14, −2.92, and −4.36 mm relative to Bregma. The extent of α-syn aggregate pathology was graded as mild to severe pathology based on the following criteria: mild, neuritic pathology or 2–3 α-syn aggregates; moderate, 4–10 α-syn aggregates; severe, over 10 α-syn aggregates seen with a 20× objective lens. Scores were determined by the observations of at least 2 mice for each group.

### 4.7. Statistics

All data were analyzed using a one-way ANOVA and multiple comparisons were analyzed with the Tukey–Kramer test. Statistical analysis was conducted using JMP Pro Version 14 (SAS institute Japan, Osaka, Japan), and the values of *p* < 0.05 were considered significant.

## Figures and Tables

**Figure 1 ijms-23-13390-f001:**
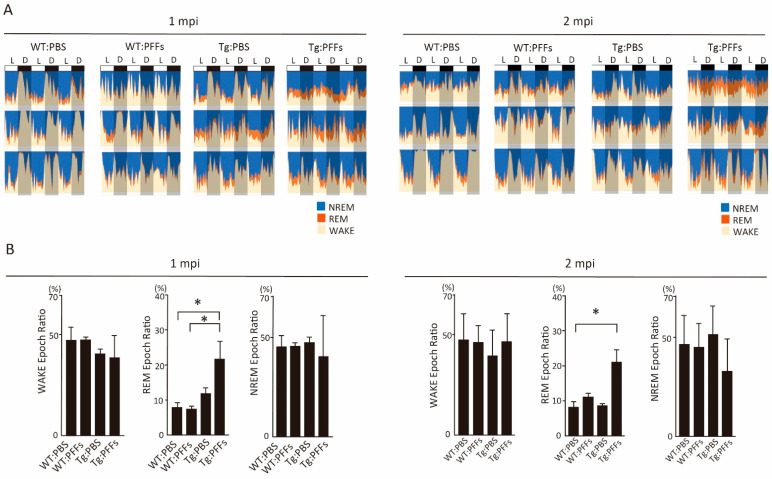
**Sleep characteristics in intrastriatal α-syn preformed fibrils (PFFs)-inoculated A53T-SNCA BAC-Tg (Tg:PFFs) mice.** (**A**) Sleep stage during every 12 h in light (L, ZT:8-20) and dark (D, ZT:20-8) phase at 1-month post inoculation (mpi) (left panel) or 2 mpi (right panel) among four groups of mice: phosphate-buffered saline (PBS)-inoculated wild-type (WT) mouse (WT:PBS), α-syn PFFs-inoculated WT mice (WT:PFFs), PBS-inoculated A53T transgenic mice (Tg:PBS), and α-syn PFFs-inoculated A53T-SNCA BAC-Tg (Tg:PFFs). The percentage of sleep stage per an hour was calculated and plotted into a timetable in the colored style. NREM: non-rapid eye movement (REM) sleep (blue), REM: REM sleep (orange), WAKE: wake (yellow). (**B**) The ratio of the number of every epoch (WAKE, REM, or NREM) to total epochs in the sleep stage at 1 mpi (left panel) or 2 mpi (right panel) among four groups of mice. Data were analyzed using a one-way ANOVA and multiple comparisons were analyzed with the Tukey–Kramer test (*n* = 4 per group). Asterisks indicate a significant difference. Bar indicates the S.D.

**Figure 2 ijms-23-13390-f002:**
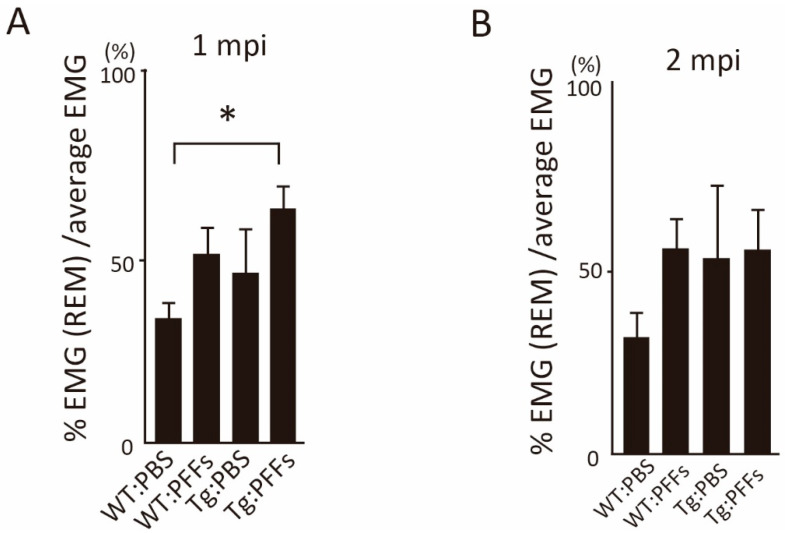
Intrastriatal α-syn PFFs inoculation aggravated the RWA phenotype in Tg:PFFs mice. (**A**,**B**) The percentage of the integral value of electromyography (EMG) per epoch of REM sleep stage relative to that of EMG in the total sleep stage at 1-month post inoculation (mpi) (**A**) or 2 mpi (**B**) among four groups of mice. Data were analyzed using a one-way ANOVA and multiple comparisons were analyzed with the Tukey–Kramer test (*n* = 4 per group). Asterisks indicate significant difference. Bar indicates the S.D.

**Figure 3 ijms-23-13390-f003:**
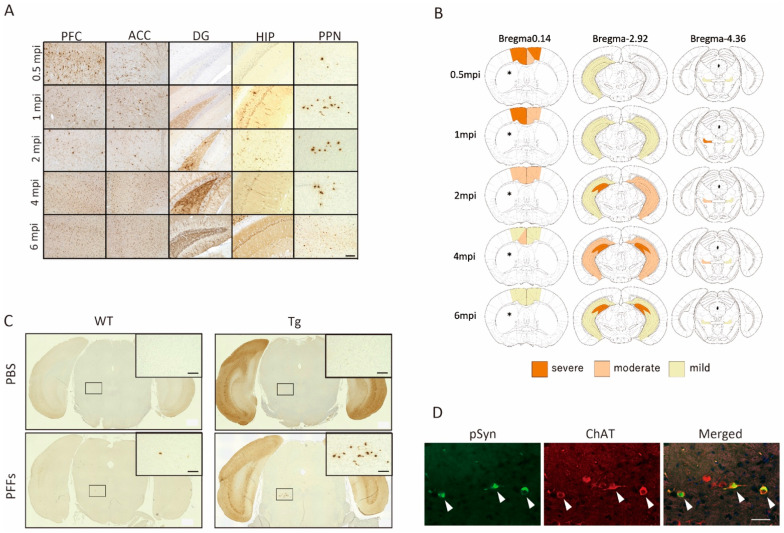
**α-syn aggregate pathologies in Tg:PFFs mice.** (**A**) P-α-syn immunostaining in various regions of the brain of α-syn PFFs-inoculated SNCA A53T transgenic (Tg) mice (Tg:PFFs mice). PFC: prefrontal cortex, ACC: anterior cingulate cortex, DG: dentate gyrus, HIP: hippocampus, PPN: pedunculopontine nucleus. Scale bar: 100 μm. (**B**) Heat map of α-syn aggregate pathologies in the Tg:PFFs mice group. * indicates the injection site of α-syn PFFs. (**C**) Phosphorylated α-synuclein (P-α-syn) immunostaining of PPN in wild-type (WT) or SNCA A53T transgenic (Tg) mice at 1-month post inoculation (1 mpi). Each selected frame contains higher magnitude image in the right-upper position. Scale bar: 100 μm. (**D**) P-α-syn (left panel, green) was co-immunostained with anti-ChAT antibody (second from left panel, red) in the PPN of the Tg:PFFs mice. Arrows indicate the co-immunostained cells. Scale bar: 100 μm.

## Data Availability

Not applicable.

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
