# Peer review of "Striatal-Inoculation of α-Synuclein Preformed Fibrils Aggravated the Phenotypes of REM Sleep without Atonia in A53T BAC-SNCA Transgenic Mice"

_ijms, 2022, doi:10.3390/ijms232113390_

Round 1

Reviewer 1 Report

The manuscript by Okuda et al. reports an interesting finding that intrastriatal inoculation of α-syn PFFs replicated the phenotype of REM sleep without atonia in A53T BAC-SNCA transgenic mice. However, there are several concerns restricting the publication of this paper.

1.     Main concern is that the manuscript has insufficient experimental data and results even as a brief report. If possible, the authors are expected to further this study and add more data to make this work more meaningful.

2.     The author claimed that quantified neck EMG activity during REM sleep was increased in Tg:PFFs mice at 1 mpi compared with WT:PBS control mice (Figure 2). The question is why compare Tg:PFFs mice with WT:PBS mice? I believe that it would be proper comparing Tg:PFF mice with Tg:PBS mice.

3.     The author claimed that the ratio of REM sleep stage was significantly increased in Tg:PFFs mice compared to Tg:PBS mice at 1 and 2 mpi (Figure 1A, B). However, the asterisks in Figure 1B seem to indicate different groups. Please check it.

4.     PPFs are mentioned many times throughout the manuscript, but there is no explanation for this abbreviation. Do you mean PFFs? Please check it.

Author Response

Response to reviewer’s Comments:

First of all, we thank the reviewers for their constructive comments. We conducted some experiments and modified the manuscript and figures. Modified descriptions in the revised manuscript are highlighted in yellow and red letters.

Reviewer#1: The manuscript by Okuda et al. reports an interesting finding that intrastriatal inoculation of α-syn PFFs replicated the phenotype of REM sleep without atonia in A53T BAC-SNCA transgenic mice. However, there are several concerns restricting the publication of this paper.

#1-1. Main concern is that the manuscript has insufficient experimental data and results even as a brief report. If possible, the authors are expected to further this study and add more data to make this work more meaningful.

  • We appreciated Reviewer#1’s important comments. We performed other behavioral analysis related to sleep and circadian rhythm (e.g., integral values of actigraphy and α-syn aggregation in orexin neurons), but we could obtain no consistent positive data other than increased RWA and increased amounts of REM sleep, both have already been observed in Tg mice in prior study (Taguchi et al.). We apologized for insufficient experimental data in this study. We instead focused on sending the simple message to the readers, ‘The sleep phenotypes were aggravated by inoculation of α-syn aggregation in A53T SNCA BAC-Tg mice.’

#1-2. The author claimed that quantified neck EMG activity during REM sleep was increased in Tg:PFFs mice at 1 mpi compared with WT:PBS control mice (Figure 2). The question is why compare Tg:PFFs mice with WT:PBS mice? I believe that it would be proper comparing Tg:PFF mice with Tg:PBS mice.

  • We appreciate Reviewer #1’s important comments. As pointed out, we compared the integral value of EMG of Tg:PFFs with Tg:PBS mice using a one-way ANOVA and multiple comparisons and there was no significant difference between them (p=31). Our study showed the significant difference in RWA phenotypes between Tg:PFFs and WT:PBS at 4 months old, which is earlier than previous study (at 11 months old between Tg and WT mice)[7], suggesting that exogenous load of α-syn PFFs could aggravate the RWA phenotypes. We added this description as the third limitation in lines 236-240 of page 7 in the Discussion section of the revised manuscript as below.

“Third, we could not find the significant differences in the sleep phenotypes between Tg:PBS and Tg:PFFs. Our study showed the significant differences in RWA phenotypes between Tg:PFFs and WT:PBS at 4 months old, which is earlier than previous study (i.e. at 11 months old between Tg and WT mice)[7], suggesting that exogenous load of α-syn PFFs could aggravate the RWA phenotypes.”

#1-3. The author claimed that the ratio of REM sleep stage was significantly increased in Tg:PFFs mice compared to Tg:PBS mice at 1 and 2 mpi (Figure 1A, B). However, the asterisks in Figure 1B seem to indicate different groups. Please check it.

  • We apologized for the typographical error. We replaced Tg:PBS with WT:PBS in line 148 on page 4 in the Result section of the revised manuscript. Thank you very much.

#1-4. PPFs are mentioned many times throughout the manuscript, but there is no explanation for this abbreviation. Do you mean PFFs? Please check it.

  • We apologize for the typographical errors. We replaced ‘PPFs’ with ‘PFFs’. Thank you very much for your important pointing out.

We thank both reviewers for their valuable comments and hope that the paper can now be accepted for publication.

Naohiro Egawa, MD, PhD

Ryosuke Takahashi, MD, PhD

Reviewer 2 Report

This article is a simple but methodologically sound study showing evidence of enhanced REM without atonia in transgenic SNCA mice injected with preformed fibrils, which occurred at 1 month post injection. Although simple, the research should be published and will be of interest given that the PFF model is currently one of the most highly used, but requires significant time investment.   The following recommendations would substantially improve the article, which I have listed as “needed” improvements and “recommended” improvements.

Needed:

1.     Please include either a picture for evidence or better description of the method used to determine that neurons contained both CHAT and pSYN. My concern is the green and red are almost exactly overlapping and the psyn channel looks very overexposed in the confocal or fluorescent picture shown- I would be concerned the CHAT staining that “co-localizes” is either bleed through or sticky secondary.  Note that every green dot is also in the red channel, suggesting a non-specific staining (There are no green dots by themselves, and this is unlikely for pSYN). The green is likely correct, but the red CHAT staining under that green may not be. Importantly, I am not saying that these data are wrong, just that the evidence presented raises my concerns that the psyn is not necessarily in CHAT positive cells. Also, please put in the figure legend where these cells are from.

2.     Add “lack of female mice” to the limitations paragraph. Given the predominance in human males, it is sufficient to justify using male mice, but the sex differences also suggest future research should be done on both sexes to better understand the disease.

Recommended:

1.     Introduction: The intro is simple and to the point, but might benefit from altering some sentences where I believe the choice of words is changing the meaning. In line 36 the author states that RBD is confirmed by presence of RWA, which is a prodromal state of PD. This I believe is correct. However, in the discussion (line 194) the author states RWA is a prodromal manifestation of isolated RBD. I think the author means to say “Isolated RWA is a prodromal manifestation of Parkinson’s disease”. This should be changed or clarified if I am misunderstanding.  The author also calls the model an RBD model, which might be accurate, but I’m not sure if that was purposeful, or if it was supposed to be a PD or synuclein model.

2.      Line 29: the first sentence should be split up- its hard to read. Also, it is not confirmed that all cells with synuclein aggregates die (they likely don’t) so unsure if this should say “dying neuronal cells”

3.     Recommend the following changes in the intro for clarity:  #43: Replace “As for the animal model” with *previous research has shown* that A53T mutated SNCA…… This helps readers understand what is old research vs. new.

4.     Recommend adding better sentence about the novelty or gap- the included sentence at line 42 “causal relationship between the severity of RWA in RBD and their underlying a-syn pathology is unclear.” This sentence makes it sound like the authors will correlate the severity of the pathology with the symptoms. Instead the authors induced a *different form of aggregation* using PFFs, rather than necessarily a more severe pathology. the figures do not show quantification of severity- the only figure showing pathological comparison is figure 3C, which does show more psyn aggregates. Figure B only shows one group (and it needs a label or mention in the legend as to which one).

Methods:

1.     No concerns.

Results: Overall methods and results are easy to read.

1)    Legends were confusing in multiple places, including use of A/B. for example in figure 1: “A.” and “B.” both have 1 mpi and 2 mpi components, so the text it is confusing when the author says “(A)” is the 1 mpi and “(B)” is the 2 mpi

Discussion

1.     There are more grammatical issues in the discussion compared to the rest of the paper- here are a few quick ones to help improve the readability.

a.     .” lines 198-199.

                                                        i.     -> change to “Changes in striatal cholinergic interneuron activity have been observed early in genetic PD rodent models.”

b.     lines 199-200.

                                                        i.     -> change to “Intrastriatal inoculation of a-syn PFFs propagates among major CNS pathways” (I think you forgot the pathway component from the reference cited here, because the sentence seems to be cut short)

c.     Sentence in lines 206-207 should start off with “A recent study….”

d.     Line 231- not sure what authors are saying.

e.     What is “focal potential activity” is this supposed to be local field potentials? (line 225). Or just activity in general?

2.     PPN cell death is mentioned in lines 229-230, but this was not examined in this study (and the sentence is confusing- why does the author mention “an attenuated intracellular signaling pathway”- is that an explanation for why the phenotype is transient. Also, from the figures it doesn’t really look like the phenotype is transient, just that the significance was lost. Do you think the phenotype is really gone 2 months post injection, at 5 months of age, when these transgenic mice develop the phenotype in prior studies? This might be interesting to discuss more in the discussion.  

Author Response

Response to reviewer’s Comments:

First of all, we thank the reviewers for their constructive comments. We conducted some experiments and modified the manuscript and figures. Modified descriptions in the revised manuscript are highlighted in yellow and red letters.

Reviewer#2: This article is a simple but methodologically sound study showing evidence of enhanced REM without atonia in transgenic SNCA mice injected with preformed fibrils, which occurred at 1 month post injection. Although simple, the research should be published and will be of interest given that the PFF model is currently one of the most highly used, but requires significant time investment.   The following recommendations would substantially improve the article, which I have listed as “needed” improvements and “recommended” improvements.

Needed:

#2-1.     Please include either a picture for evidence or better description of the method used to determine that neurons contained both CHAT and pSYN. My concern is the green and red are almost exactly overlapping and the psyn channel looks very overexposed in the confocal or fluorescent picture shown- I would be concerned the CHAT staining that “co-localizes” is either bleed through or sticky secondary.  Note that every green dot is also in the red channel, suggesting a non-specific staining (There are no green dots by themselves, and this is unlikely for pSYN). The green is likely correct, but the red CHAT staining under that green may not be. Importantly, I am not saying that these data are wrong, just that the evidence presented raises my concerns that the psyn is not necessarily in CHAT positive cells. Also, please put in the figure legend where these cells are from.

  • We appreciate Reviewer#2’s important comments. We changed the antibody ChAT (Ab144p) and performed the co-immunostaining with ChAT and p-α-syn Almost all p-α-syn were co-localized with ChAT-expressing cells, while not all ChAT-expressing cells showed p-α-syn positive cells. We added the location of the staining. We showed the revised pictures and new figure 3D as below(please attached word file). Thank you for your constructive comments.

#2-2.     Add “lack of female mice” to the limitations paragraph. Given the predominance in human males, it is sufficient to justify using male mice, but the sex differences also suggest future research should be done on both sexes to better understand the disease.

  • We appreciate Reviewer#2’s important comments. Since RBD is generally more common in male than in female, we used male mice to recapitulate the character of RBD. In addition, we usually use male mice for sleep analysis to avoid the effects of estrus cycle (e.g., Chari T et al., PMID: 32714163; Dib R et al. PMID: 34195482). The previous study analyzing WT mice injected with α-syn PFFs into the CNS reported that male PFF-injected mice developed more severe α-syn pathology than female ones [Mason DM et al. PMID:30854742], while there is no significant difference in REM sleep amounts between sexes [Dib R et al. PMID: 34195482]. We agree with the importance of sex differences in diseases and the research. We added the following descriptions as the fourth limitation by citing the Mason et al. and Dib et al. papers in line 240-246 on page 7 in the Discussion of the revised manuscript.

“Fourth, we only used male mice for sleep analysis in the present study since RBD is generally more common in male than in female. The previous study analyzing WT mice inoculated with α-syn PFFs into the CNS reported that male PFF-injected mice developed more severe α-syn pathology than female ones[22], while there is no significant difference in REM sleep amounts between the sexes[23]. Future research should be done on both sexes to understand the disease in future studies.”

Recommended:

#2-3.     Introduction: The intro is simple and to the point, but might benefit from altering some sentences where I believe the choice of words is changing the meaning. In line 36 the author states that RBD is confirmed by presence of RWA, which is a prodromal state of PD. This I believe is correct. However, in the discussion (line 194) the author states RWA is a prodromal manifestation of isolated RBD. I think the author means to say “Isolated RWA is a prodromal manifestation of Parkinson’s disease”. This should be changed or clarified if I am misunderstanding.  The author also calls the model an RBD model, which might be accurate, but I’m not sure if that was purposeful, or if it was supposed to be a PD or synuclein model.

  • We appreciate Reviewer#2’s great comments. We meant “Isolated RWA precedes RBD and worsens with PD progression” as reference [15]. We clarified and corrected it in lines 199 on page 6 in the Discussion section of the revised manuscript as below(please see the attached word file).

‘Isolated RWA precedes RBD and worsens with PD progression, which imply the underlying progressive process of α-syn pathology in PD.’

#2-4.      Line 29: the first sentence should be split up- its hard to read. Also, it is not confirmed that all cells with synuclein aggregates die (they likely don’t) so unsure if this should say “dying neuronal cells”

  • We appreciate Reviewer#2’s important comments. We split up the long sentence and modified the description in line 29-30 on page 1 in the Introduction section of the revised manuscript as bellow.

“Synucleinopathy represented by Parkinson’s disease (PD) is an entity of neurodegenerative diseases, which defining pathological feature is the presence of Lewy bodies in the brain. Lewy bodies are predominantly composed of alpha synuclein (α-syn) aggregation.”

#2-5.     Recommend the following changes in the intro for clarity:  #43: Replace “As for the animal model” with *previous research has shown* that A53T mutated SNCA…… This helps readers understand what is old research vs. new.

  • We appreciate Reviewer#2’s great comments. As pointed out, we replace it in line 45-46 on page 2 in the Introduction section of the revised manuscript as bellow.

Previous research has shown that, A53T mutated SNCA bacterial artificial chromosome transgenic (A53T SNCA BAC-Tg) mice exhibited hyposmia and the RBD-like phenotype at five months of age

#2-6.     Recommend adding better sentence about the novelty or gap- the included sentence at line 42 “causal relationship between the severity of RWA in RBD and their underlying a-syn pathology is unclear.” This sentence makes it sound like the authors will correlate the severity of the pathology with the symptoms. Instead the authors induced a *different form of aggregation* using PFFs, rather than necessarily a more severe pathology. the figures do not show quantification of severity- the only figure showing pathological comparison is figure 3C, which does show more psyn aggregates. Figure B only shows one group (and it needs a label or mention in the legend as to which one).

  • We appreciate Reviewer#2’s important comments. As pointed out, we did not examine the severity of α-syn (the quantification of α-syn) in our study and the description should be misleading. To clarify the gap and the novelty in our study, we corrected the description in lines 43-44 on page 1 in the Introduction section of the revised manuscript as below.

“The causal relationship between RWA in RBD and the exogenous load of α-syn aggregation is unclear.”

In addition, we clarified the group in Figure 3B legend as below. Thank you very much.

“Heat map of α-syn aggregates pathology in the Tg:PFFs mice group.”

Results: Overall methods and results are easy to read.

#2-7.  Legends were confusing in multiple places, including use of A/B. for example in figure 1: “A.” and “B.” both have 1 mpi and 2 mpi components, so the text it is confusing when the author says “(A)” is the 1 mpi and “(B)” is the 2 mpi.

  • We apologized for our errors. We corrected the description in lines 153-159 on page 4 in the Figure legends of the revised manuscript as below. Thank you very much for your important pointing out.

Discussion

#2-8. There are more grammatical issues in the discussion compared to the rest of the paper- here are a few quick ones to help improve the readability.

  1. .” lines 198-199.   -> change to “Changes in striatal cholinergic interneuron activity have been observed early in genetic PD rodent models.”
  • We apologized for our error. We corrected the description in line 204 on page 6 in the Discussion section of the revised manuscript. Thank you very much.

  1. lines 199-200.   -> change to “Intrastriatal inoculation of a-syn PFFs propagates among major CNS pathways” (I think you forgot the pathway component from the reference cited here, because the sentence seems to be cut short)
  • We apologize for the error. We added ’pathways’ in the sentence in line 205 on page 6 as below. Thank you very much.

“Intrastriatal inoculation of α-syn PFFs propagates among major central nervous system (CNS) pathways”

  1. Sentence in lines 206-207 should start off with “A recent study….”
  • We apologize for the error. We corrected and started off with ’A recent study...’ in the sentence in line 212 on page 6. Thank you very much.

  1. Line 231- not sure what authors are saying.
  • We apologize for the misleading description. We omitted the sentence. Thank you very much.

  1. What is “focal potential activity” is this supposed to be local field potentials? (line 225). Or just activity in general?
  • We corrected the word in line 230 on page 7 in the Discussion section of the revised manuscript. Thank you very much.

#2-9.     PPN cell death is mentioned in lines 229-230, but this was not examined in this study (and the sentence is confusing- why does the author mention “an attenuated intracellular signaling pathway”- is that an explanation for why the phenotype is transient. Also, from the figures it doesn’t really look like the phenotype is transient, just that the significance was lost. Do you think the phenotype is really gone 2 months post injection, at 5 months of age, when these transgenic mice develop the phenotype in prior studies? This might be interesting to discuss more in the discussion.  

  • We appreciate Reviewer#2’s important comments. As pointed out, we did not examine cell death. Although the differences in the sleep and RWA phenotypes among groups was not significant, the tendency of the increased amount of REM sleep and RWA in Tg:PFFs mice was constant at 2 mpi (i.e. 5 months of age). We omitted the misleading sentences and instead added the description as to the limitation that we could not find the difference in the phenotypes between Tg:PBS and Tg:PFFs in line 236-240 on page 7 in the Discussion section as below.

“Third, we could not find the significant differences in the sleep phenotypes between Tg:PBS and Tg:PFFs. Our study showed the significant differences in RWA phenotypes between Tg:PFFs and WT:PBS at 4 months old, which is earlier than previous study (i.e. 11 months old between Tg and WT mice)[7], suggesting that exogenous load of α-syn PFFs could aggravate the RWA phenotypes.”

We thank both reviewers for their valuable comments and hope that the paper can now be accepted for publication.

Naohiro Egawa, MD, PhD

Ryosuke Takahashi, MD, PhD

Round 2

Reviewer 1 Report

After careful consideration, given that the interesting findings in this paper could further more valuable studies, the manuscript is recommended to be published in its current form as a brief report.